# Novel targets and potential therapeutic approaches for chromoblastomycosis: phosphatase/peptidase inhibitors and metal-based compounds containing 1,10-phenanthroline

**Ingrid de Souza e Sousa¹, Marcela Queiroz Granato¹, Vanila Faber Palmeira², André Luis Souza dos Santos²,³, Lucimar Ferreira Kneipp¹,³/⁺**

¹Fundação Oswaldo Cruz-Fiocruz, Instituto Oswaldo Cruz, Laboratório de Taxonomia, Bioquímica e Bioprospecção de Fungos, Rio de Janeiro, RJ, Brasil
²Universidade Federal do Rio de Janeiro, Instituto de Microbiologia Paulo de Góes, Laboratório de Estudos Avançados de Microrganismos Emergentes e Resistentes, Rio de Janeiro, RJ, Brasil
³Fundação de Amparo à Pesquisa do Estado do Rio de Janeiro, Rede Micologia RJ, Rio de Janeiro, RJ, Brasil

Chromoblastomycosis (CBM) is a chronic, debilitating subcutaneous mycosis that remains a major therapeutic challenge due to its limited responsiveness to conventional antifungal agents. Our research group investigated key cellular mechanisms underlying fungal virulence, persistence, and resistance, revealing that CBM-associated fungi can exhibit ectophosphatase and calcineurin activities, secrete aspartic and metallo-type peptidases, and form highly structured biofilms that reinforce their chronic behaviour and tolerance to treatment. Targeted inhibition of these enzymatic systems using classical inhibitors of peptidases [*e.g.*, human immunodeficiency virus (HIV) aspartic peptidase inhibitors], acid phosphatases (*e.g.*, sodium orthovanadate), and calcineurin (*e.g.*, tacrolimus and cyclosporine A) markedly impaired fungal growth, morphogenesis, biofilm development, and/or host-cell interactions, underscoring their potential roles in key fungal biological processes and infection establishment. Furthermore, coordination compounds incorporating transition metals (*e.g.*, silver) and 1,10-phenanthroline-derived ligands demonstrated potent antifungal efficacy against CBM-associated fungi and may interfere with key physiological and virulence-associated pathways. Collectively, these findings advance the understanding of CBM fungal pathophysiology, unveiling novel molecular targets and highlighting opportunities for alternative therapeutic strategies and antifungal drug development.

Key words: coordination compounds - HIV peptidase inhibitors - calcineurin inhibitors - antifungal activity - cellular interaction - biofilm

Chromoblastomycosis (CBM) is an implantation mycosis considered a neglected tropical disease by the World Health Organisation (WHO) since 2017.[1] CBM is worldwide and affects different workers/individuals from rural to urban centres, when they come into contact with soil and decaying wood, where most fungi live in nature.[2,3] This disease is caused by several fungi belonging to the *Fonsecaea*, *Phialophora*, *Cladophialophora*, *Rhinocladiella*, *Exophiala* and *Veronaea* genera.[4] They are melanised filamentous fungi that exhibit a parasitic form known as sclerotic cells, which are characteristic of CBM and confer increased resistance to both immune defences and antifungal treatments.[5,6] Depending on the etiological agents and immune status of the host, infections can be limited in cutaneous and subcutaneous tissues or cause systemic commitment including endophthalmitis, meningoencephalitis and even other diseases such as mycetoma and phaeohyphomycosis.[4,5] CBM complications are associated with chronic lymphedema that can develop to elephantiasis, bacterial infections and malignant transformation into squamous cell carcinoma.[7] This mycosis is also recognised as an occupational disease and can incapacitate individuals affected by the causative fungi.[5]

Despite the significant public health impact of CBM, particularly affecting low-income communities in tropical regions, the disease still lacks a specific treatment approved by the Food and Drug Administration (FDA) or any other regulatory authority. The treatment of fungal infections continues to pose significant challenges, primarily due to the narrow spectrum of available antifungals, their potential adverse effects, and the rising incidence of antifungal resistance (for review see Li et al.[8]) Treatment of CBM is particularly difficult; although classified as a neglected tropical disease by the WHO, it was not included in the 2022 fungal priority pathogens list.[9] To date, its clinical treatment consists of multiple approaches, depending on the size, location, and duration of CBM lesions, as well as the immunological status of the host. Treatment includes physical methods such as surgery, thermotherapies (such as cryotherapy), laser and photodynamic therapies, in addition to antifungal

**doi:** 10.1590/0074-02760250305
**Financial support:** CNPq, FAPERJ, CAPES (financial code 001), FIOCRUZ.
**+ Corresponding author:** lucimar@ioc.fiocruz.br | ⓘ https://orcid.org/0000-0002-3507-4689

Handling editor: Ana Carolina Paulo Vicente | ⓘ https://orcid.org/0000-0001-7086-2042

therapies.[5] Overall, conventional antifungal agents used are itraconazole that can be combined with either 5-flucytosine or terbinafine in patients presenting refractory disease. In severe and disseminated cases, amphotericin B can be prescribed.[5,10] Current CBM treatments have limitations, as they are long-lasting, associated with side effects, and some fungal species have developed resistance.[4] Besides immunotherapeutic strategies, several studies have also proposed natural and synthetic compounds for the alternative treatment of CBM.[11,12,13,14]

The pathogenic mechanisms involved in CBM are not yet well established. However, some studies, including those conducted by our group, have identified important molecules and structures, either associated with the cell wall or secreted by *Fonsecaea pedrosoi*, the main etiological agent of CBM,[15-34] and have correlated them with pathophysiological events (Table). Studies from our group, derived from a sustained line of investigation, have contributed to the understanding of key cellular and biological events in CBM-causing fungi. These findings form the basis of the present review, which integrates them into a unified perspective and provides a framework to guide future studies, focusing on the identification of novel targets and potential therapeutic strategies for this neglected and debilitating infection. Beyond *F. pedrosoi*, our studies also revealed that *Rhinocladiella aquaspersa* exhibits ectophosphatase activity, whereas *Phialophora verrucosa* secretes aspartic and metallopeptidases and forms biofilms.[33,35,36,37] These findings, including the structural and enzymatic features described in *F. pedrosoi*, *R. aquaspersa*, and *P. verrucosa*, as well as the effects of phosphatase and peptidase inhibitors and/or 1,10-phenanthroline (phen)-derived compounds on these fungal cells, are discussed in detail in this review.

## Phosphatases and phosphatase inhibitors

Protein phosphorylation is the highly prevalent post-translational modification, regulating most cellular proteins either directly or indirectly. In eukaryotic cells, phosphorylation typically occurs on serine, threonine and tyrosine residues, often leading to conformational changes that alter protein function.[38,39] This process is controlled by the opposing actions of protein kinases, which add phosphate groups, and phosphatases, which remove them regulating crucial cellular functions.[40,41] Phosphatases are classified according to their mode of action as serine/threonine or tyrosine phosphatases, and based on their cellular location as intracellular, secreted/extracellular, or cell wall-associated.[41,42] Pathological conditions linked to phosphorylation imbalance include cancer and inflammation-related disorders.[42] Therapies that target protein kinases and phosphatases remain challenging due to their conserved structures, complex regulation, and limited drug specificity.[43,44] However, efforts to design allosteric inhibitors and small-molecule inhibitors targeting phosphatases, as monotherapy and in combination with other agents, have led to several candidates progressing to clinical trials.[44] In fungi, several studies including systematic functional phosphatome have shown that phosphatases play key roles

TABLE
Molecules and structures of *Fonsecaea pedrosoi* identified as potential antifungal targets

| Cellular structures/antifungal targets | Main potential functions | References |
|---|---|---|
| Melanin | Resistance against macrophage action<br>Induction of the immune response<br>Protecting fungal against oxidative stress<br>Inhibition of nitric oxide production by macrophages<br>Human complement system activation | [15,16,17,18,19] |
| Sialic acid/Sialylglycoconjugates | Protection against immune destruction by host cells | [20] |
| Lectin-like surface adhesin | Adhesion to host cells<br>Fungal internalisation | [21] |
| Cell-wall phosphatase (ectophosphatase) | Adhesion to host cell | [22,23] |
| Glucosylceramide | Fungal growth<br>Resistance against macrophage action | [24,25] |
| Secreted metallopeptidase | Fungal growth<br>Fungal dissemination | [26] |
| Secreted aspartic peptidase | Fungal growth<br>Adhesion to host cells<br>Fungal dissemination | [27,28,29,30] |
| Lipases (Phospholipase and esterase) | Host cell penetration | [31] |
| Extracellular vesicles | Macrophages cytokine modulation | [32] |
| Biofilm | Antifungal resistance | [33] |
| Intracellular phosphatase (Calcineurin) | Fungal growth<br>Filamentation | [34] |

in signalling pathways to maintaining cellular homoe-ostasis. They are involved in essential fungal biological processes such as cell wall synthesis, maintaining cellular integrity under stress conditions, hyphal formation, cell differentiation and in events associated with fungal pathogenicity.[40,41,45,46,47,48]

Our group described the presence of cell wall-associated phosphatases in CBM-causing fungi *F. pedrosoi* and *R. aquaspersa*.[22,23,35] Theses enzymes, also called ectophosphatases, have their catalytic site facing the extracellular environment, and their cell wall localisation was confirmed in *F. pedrosoi* (conidia and mycelia) after visualisation of electron-dense cerium phosphate deposits using transmission electron microscopy.[22,23] In addition, we showed using biochemical assays that both fungi were able to hydrolyse phosphorylated substrates, such as *p*-nitrophenylphosphate and phosphorylated amino acids including serine, threonine and tyrosine. *F. pedrosoi* and *R. aquaspersa* phosphatase activities reached a maximum at acidic pH, which was confirmed after treatment with classical acid phosphatase inhibitors such as sodium salts of molybdate, orthovanadate and fluoride that highly reduced their activities.[22,35] It is well-known that phosphatase is modulated by different cations.[41] Consistently, we showed that the acid phosphatase of *R. aquaspersa* conidia was stimulated by cobalt ions while *F. pedrosoi* mycelia and conidia were modulated positively by zinc and ferric, respectively.[22,23,35]

Our data revealed that *F. pedrosoi* freshly isolated from a human case of CBM showed increased enzyme activity than laboratory-adapted one, suggesting that the surface phosphatase might be modulated by interaction with the host cells.[22,35] Considering the lack of phosphatase-deficient mutants of these fungi, we performed experiments to produce fungal cells with different levels of phosphatase. The regulation of phosphatases by phosphate in culture medium had already been reported in other fungi, for instance *Aspergillus niger* and *Saccharomyces cerevisiae*.[49,50] As ectophosphatase activities of *F. pedrosoi* and *R. aquaspersa* conidia were strongly inhibited by inorganic phosphate, we cultivated both fungi in culture medium without exogenous inorganic phosphate addition. In this condition, we showed that the enzymatic activity of both fungal cells was highly increased by around 100-fold than those conidia that were grown in the culture medium containing inorganic phosphate with a basal phosphatase activity.[23,35] Interestingly, conidia of *F. pedrosoi* and *R. aquaspersa* with high ectophosphatase activity had a greater adhesion level to fibroblasts 3T3-L1 cell line, epithelial cell from the kidney of an African green monkey (MA-104) and Chinese ovary hamster (CHO) cells, respectively. This adhesion was inhibited when both fungi were pretreated with sodium orthovanadate, an irreversible inhibitor of both ectophosphatase activities.[23,35] In *R. aquaspersa*, host cell adherence was also significantly reduced after treatment with sodium molybdate and anti-phosphatase antibodies. These findings suggest that this surface enzyme, identified in both CBM-causing fungi, may contribute to fungus-host interactions.[23,35] The involvement of ectophosphatases in cellular interactions is not well

established; however, several hypotheses have been proposed, including: (i) removal of inorganic phosphate from the host cell surface may expose other molecules or structures that facilitate fungal attachment; (ii) ectophosphatases may contain structural domains that function as adhesins, thereby contributing to fungal adhesion to host cells; and (iii) dephosphorylation of host cell surface molecules may reduce electrostatic repulsion, favouring fungal binding to host cells.[41,51] Notably, studies have shown that fungal ectophosphatases are relevant in host interaction events. As described for CBM-causing fungi, the inhibition of this enzyme by sodium orthovanadate also decreased fungal attachment to epithelial cells in other pathogenic fungi, including *Cryptococcus neoformans*, *Candida parapsilosis*, and *Candida albicans*.[52,53,54] In addition, Portela et al.[54] showed that *C. albicans* isolates from human immunodeficiency virus (HIV)-positive children presented greater ectophosphatase activity and adherence to MA-104 epithelial cells than yeasts from HIV negative patients. This activity inhibition by orthovanadate reduced levels of interaction corroborating that ectophosphatase may contribute to the early mechanisms required for recognition between fungal and host cells.

Moreover, the putative role of this enzyme in the *F. pedrosoi* and *R. aquaspersa* parasitism was also discussed, since the morphotypes of these fungi showed clearly distinct levels of activities.[22,35] The profile was completely different, for *R. aquaspersa* conidial forms showed higher activity than mycelia and sclerotic cells, while for *F. pedrosoi* the parasitic form sclerotic cells presented the greater activity following by mycelia and conidia.[22,35] In addition, we showed that differentiation inducers, such as propranolol and platelet-activating factors, were able to stimulate the activity of *F. pedrosoi* ectophosphatase, supporting a possible association with cell differentiation.[55] Beyond their potential roles in adhesion and virulence, fungal ectophosphatases may also participate in microbial nutrition by providing a source of inorganic phosphate through the hydrolysis of organic phosphate compounds.[41] However, we observed that both acid phosphatase inhibitors tested (sodium orthovanadate and sodium molybdate) were unable to impar *F. pedrosoi* growth (unpublished data) and slightly inhibited the proliferation of *R. aquaspersa* only at high (5 mM) concentration.[35] Undoubtedly, ectophosphatase remains a potential antifungal target, since in fungi, including *F. pedrosoi* and *R. aquaspersa*, it can be involved in early events responsible for disease establishment. Further investigations into the structural and functional differences between fungal and host phosphatases are essential for the future design of selective inhibitors targeting fungal infections.

Besides cell wall-phosphatases, intracellular phosphatases as calcineurin are also an attractive antifungal target.[56,57,58] Calcineurin is a calcium/calmodulin-activated protein phosphatase and a key regulator of cellular stress responses in all eukaryotes. This enzyme is formed by a catalytic A subunit and a regulatory B subunit, both crucial for its activity. Upon calcium influx into the cytosol, calcineurin binds to calcium-calmodulin,

triggering a conformational change that displaces its autoinhibitory domain and activates the catalytic site responsible for dephosphorylating protein substrates.[59] In pathogenic fungi, it controls numerous physiological processes, including cell cycle progression, morphogenesis, responses to antifungal drugs and virulence.[58] The regulation of calcineurin is a complex and highly conserved process that involves multiple levels of control, ranging from activation by cellular signals to modulation by regulatory proteins and inhibitors.[60] The calcineurin inhibitors commonly used in clinical practice are natural products such as tacrolimus (known also as FK506) and cyclosporine A. Tacrolimus binds to FK506-binding protein 12 (FKBP12), while cyclosporine A associates with cyclophilin A; the resulting protein-drug complexes interact with calcineurin and inhibit its function.[57,60] Studies have shown that these inhibitory assemblies block calcineurin activity in pathogenic fungi including *C. neoformans*, *Aspergillus fumigatus*, *C. albicans*, and *Candidozyma auris* (formerly *Candida auris*) making it a promising antifungal target, as its inhibition is associated with crucial outcomes, including reduced virulence and increased susceptibility to antifungal agents.[57,61,62,63,64,65]

Recently, we showed that *F. pedrosoi* conidia exhibit an intracellular enzyme capable of hydrolysing the artificial substrate *p*-nitrophenylphosphate, with its activity enhanced by calcium chloride and calmodulin, an activator of calcineurin, as well as inhibited by tacrolimus, cyclosporine A, and the calcium chelator ethylene glycol tetra-acetic acid.[34] These findings were corroborated by the recognition of the fungal extract by an anti-calcineurin antibody, and by the reduction in fluorescence following treatment with tacrolimus and cyclosporine A, as measured by flow cytometry.[34] In addition, our results showed that the both calcineurin inhibitors markedly reduced fungal viability, after propidium iodide staining using flow cytometry, and also inhibited fungal proliferation according to Clinical and Laboratory Standards Institute (CLSI) antifungal susceptibility testing. Morphological damage was confirmed by transmission electron microscopy, which revealed cytoplasmic condensation, extensive intracellular vacuolisation, and cellular disruption in the fungal cells.[34] Several studies have shown that calcineurin inhibitors can promote synergistic effects with antifungal agents.[64,66,67,68] However, the growth of some fungi is affected only when calcineurin inhibitors are combined with other antifungal agents.[68] Our findings showed that tacrolimus and cyclosporine A inhibit conidial growth either alone or in combination with itraconazole, exhibiting a synergistic action. Additionally, our data revealed that treatment of conidia with subinhibitory concentrations of tacrolimus and cyclosporine A inhibited the transformation of *F. pedrosoi* conidia into mycelia.[34] Calcineurin inhibitors similarly affected hyphal formation in other fungi, including *Candida tropicalis*, *A. fumigatus,* and *Trichosporon asahii*.[69,70,71] Evidence indicates that inhibiting this process is critical for reducing fungal viability and infection, since filamentation is essential for both nutrient acquisition and early stages of host invasion and dissemination.[72]

Calcineurin inhibitors, commonly used in clinical practice to prevent organ transplant rejection and treat immune-mediated disorders, are immunosuppressive agents.[73] For this reason, they currently represent a risk factor and are therefore not recommended for the treatment of fungal infections. However, experimental studies have shown that analogues of calcineurin inhibitors exhibit fungicidal activity with little or no immunosuppressive effects and may represent a promising strategy for the development of novel antifungal therapies.[74,75,76] An alternative approach to improve the antifungal activity of tacrolimus involves combining it with antagonists that compete for FKBP12 binding and selectively permeate human, but not fungal cells, maintaining efficacy while minimising immunosuppressive effects.[77] Therefore, promoting research on fungal-selective calcineurin inhibitors is essential for the development of targeted therapies against fungal infections.

## Peptidases and peptidase inhibitors

Another group of fungal enzymes involved in a wide range of biological processes, including nutrition, growth, differentiation, and host cell interaction, are peptidases.[78,79,80] These enzymes catalyse and cleave protein peptide bonds, resulting in free peptides or amino acids. Depending on the evolutionary relationships and amino acid sequences of peptidases, they are categorised into different clans and families.[81] According to their phylogenetic relationships and mechanisms of action, proteolytic enzymes are classified into different groups, including asparagine-, aspartic-, cysteine-, glutamic-, metallo-, threonine-, and serine-peptidases.[81,82] Peptidase regulatory strategies involve inhibitors that interact with the catalytic domain of the enzyme to suppress proteolytic activity. Peptidase inhibitors are grouped into different classes according to their mechanisms of action;[83] however, this review focuses specifically on aspartic peptidase and metallopeptidase inhibitors. Aspartic peptidase inhibitors are best known for their role in highly active antiretroviral therapy, the most effective treatment for acquired immunodeficiency syndrome. Several HIV-peptidase inhibitors (HIV-PIs) have been approved by the FDA, including saquinavir, ritonavir, indinavir, nelfinavir, amprenavir, lopinavir, and tipranavir.[84] In contrast, metalloproteinase inhibitors, such as angiotensin-converting enzyme inhibitors, are widely used to treat hypertension and heart failure.[85] These examples highlight how different classes of peptidase inhibitors play important roles in both physiological regulation and therapeutic interventions for a range of conditions, including infectious diseases.[84]

Interestingly, clinical trials demonstrated that the introduction of HIV-PIs in the treatment of HIV-infected individuals led to a significant reduction in invasive fungal infections.[86] Experimental studies revealed that HIV-PIs, originally developed to target the HIV aspartic peptidase, effectively block fungal aspartic peptidases, thereby disrupting key biological processes such as cellular growth, metabolic activity, morphogenesis, adhesion, and biofilm formation.[87,88,89] Notably, indinavir and ritonavir showed therapeutic efficacy in experimen-

tal models of vaginal candidiasis, with outcomes comparable to fluconazole.[90] In a murine model of cryptococcosis, tipranavir reduced fungal burden in the brain and liver of mice infected with *C. neoformans*.[91] Additionally, lopinavir impaired fungal adhesion to epithelial cells and reduced infection severity in both immunocompetent and immunosuppressed mice.[88] These findings highlight fungal peptidases as promising drug targets, offering novel therapeutic mechanisms that differ from those of conventional antifungal agents.

For many years, studies based on enzymatic profiles of CBM-associated fungi were used for the taxonomic differentiation of related species. Consequently, several clinical and environmental isolates of *F. pedrosoi*, for example, were classified as non-peptidase-producing fungi.[92,93] These studies were based primarily on the detection of clear halos around fungal colonies grown on solid media containing proteinaceous substrates such as gelatine, bovine serum albumin, and/or casein. This approach represents a simple method for peptidase detection and is mainly suitable for enzymes exhibiting high proteolytic activity capable of generating visible halos.[82] Our studies, employing more sensitive approaches such as concentrated culture supernatants, chromogenic/fluorogenic substrates, and selective inhibitors, provided evidence that CBM-causing fungi, including *F. pedrosoi* and *P. verrucosa*, are capable of secreting peptidases.[26-30,36,37] We demonstrated that different morphotypes of *F. pedrosoi* (conidia, mycelia, and sclerotic cells) are capable of secreting aspartic peptidases with optimal activity at acidic pH (2.0 - 4.0), exhibiting distinct characteristics in their ability to hydrolyse proteinaceous substrates and in sensitivity to peptidase inhibitors. Under the conditions used and based on soluble protein cleavage profiles, both *F. pedrosoi* conidia and mycelia were able to hydrolyse human or bovine serum albumin and extracellular matrix proteins, such as human laminin and fibronectin, whereas sclerotic cells did not degrade fibronectin.[26,27,30] In line with these observations, the culture supernatant of *P. verrucosa* conidia hydrolysed human serum albumin, and aspartic peptidase activity in both fungi was supported by the substantial reduction in enzymatic activity following treatment with the classical inhibitor pepstatin A and HIV-PIs. Among HIV-PIs, distinct activity profiles were observed, with lopinavir and ritonavir being the most effective inhibitors of the enzyme secreted by *P. verrucosa* (> 75%), whereas nelfinavir and indinavir most efficiently inhibited (> 70%) the peptidase activity of *F. pedrosoi* conidia, mycelia, and sclerotic cells.[27,28,30,37] Our findings support the notion that pathogenic fungi secrete peptidases capable of degrading serum and extracellular matrix proteins, which may contribute to their ability to circumvent natural human barriers and host defences.[94]

Furthermore, HIV-PIs produced a direct impact on *P. verrucosa* and *F. pedrosoi* viability and growth. Among them, ritonavir inhibited (> 40-60%) *P. verrucosa* growth in a dose-dependent manner, while nelfinavir showed the highest antifungal activity inhibiting > 90% the growth of *F. pedrosoi* conidia and sclerotic cells.[30,37] Notably, the secretion of these enzymes can be influenced by fungal morphological form and environmental conditions, as described in *C. albicans* and *Paracoccidioides brasiliensis*.[95,96] This modulation of peptidase secretion in *F. pedrosoi* morphotypes may be associated with fungal adaptation to the human host, potentially contributing to replication in different tissues.[97] The impact of HIV-PIs on fungal viability and growth was confirmed by irreversible ultrastructural alterations in both fungi.[28,29,37] Importantly, scanning electron microscopy analyses showed that treatment of *P. verrucosa* with nelfinavir, lopinavir and ritonavir induced various cellular changes, including surface deposits, cell wall damage, and cytolysis.[37] Moreover, transmission electron microscopy revealed that nelfinavir and saquinavir affected *F. pedrosoi* conidia morphology causing invagination of the cell membrane, disorganisation of the cell wall and an increase in cytoplasmic vacuoles.[28] Blocking aspartic peptidases can impair the ability of fungal cells to acquire peptides and amino acids, directly affecting fungal growth. Notably, HIV-PIs may exert multifactorial effects that disrupt fungal homeostasis and ultimately lead to cell death.[98]

As the ability to survive and replicate within host cells is a key determinant of fungal virulence,[99] we examined the impact of HIV-PIs on the adhesion of *F. pedrosoi* and *P. verrucosa* conidia to animal cells. Saquinavir was the most potent against adhesion and endocytosis of *F. pedrosoi* conidia and CHO epithelial cells.[28] The interaction conidia-CHO cells promoted a high number of hyphae that was reverted mainly after saquinavir treatment indicating that morphological transition was induced by contact with animal cells. Moreover, we evaluated the effect of HIV-PIs on the *F. pedrosoi* conidia using 3T3-L1 mouse fibroblasts and murine macrophage cell line RAW 264.7.[28] For fibroblasts, nelfinavir, saquinavir and also indinavir markedly inhibited mainly the fungal endocytic index (~80%), whereas in macrophages, both adhesion and endocytic indices were reduced by approximately 50% after treatment with saquinavir and indinavir. Our data demonstrated that lopinavir and ritonavir, either alone or in combination, were effective in inhibiting *P. verrucosa* conidial adhesion to human macrophages derived the acute monocytic leukaemia cell line THP-1.[37] Additionally, we investigated the killing capability of macrophages cells against *F. pedrosoi* and *P. verrucosa* conidia treated with HIV-PIs. Our data revealed that mainly indinavir and nelfinavir were able to reduce the intracellular conidia viability of *F. pedrosoi*, while lopinavir combined with ritonavir decreased viable number of *P. verrucosa* conidia,[28,37] suggesting that these inhibitors can also affect the microbicidal capacity of macrophages. The HIV-PIs susceptibility of both fungi to animal cells may also be linked to mechanisms of antimicrobial immunity in phagocytic cells, such as the enhanced oxidative burst reported in other fungal infections.[100]

To better understand the effects of aspartic peptidase inhibitors on morphological changes and increased susceptibility to phagocytosis in *F. pedrosoi* conidia, we examined the ability of HIV-PIs to interfere with the expression of conidial surface molecules associated with virulence, including glycoconjugates containing mannose, sialic acid, and glucosylceramide. The effects

of HIV-PIs were also investigated on *F. pedrosoi* sterol, melanin, phospholipase and esterase.[29] Indinavir was the most effective in reducing the amount of mannose-rich glycoconjugates on the surface of conidia, and these results may partly explain the reduction in conidia adhesion to mammalian cells.[28,29] The interfering of HIV-PI in the expression of virulence-associated surface molecules was also reported by *C. albicans*.[88] On the other hand, treatment with HIV-PIs did not alter the expression of sialic acid on the surface glycoconjugates of *F. pedrosoi* conidia under the conditions tested. Saquinavir and ritonavir were more effective in reducing ergosterol levels, a key lipid involved in maintaining the fluidity, permeability, and structural integrity of the fungal plasma membrane and a well-recognised target of classical antifungal drugs.[101] Such inhibition may alter fungal membrane composition and contribute to the deleterious effects observed with HIV-PIs. Although the precise mechanism of action remains unclear, the antifungal activity of HIV-PIs may be mediated, at least in part, by ergosterol depletion, which could indirectly impair membrane-associated processes and other virulence-related pathways.

Treatment of conidia with HIV-PIs was also associated with reduced melanin production, a major virulence factor of dematiaceous fungi, with indinavir and ritonavir showing the most pronounced effects.[43] These findings are promising, as melanin is known to protect fungal cells, including CBM-causing fungi, against host immune response and the action of antimicrobial agents.[102,103] These results suggest that reduced melanin expression may have contributed to increased susceptibility of conidia to macrophage-mediated killing after treatment with HIV-PIs.[27,28] In addition, increased recognition of anti-glucosylceramide antibodies on the surface of *F. pedrosoi* conidial cells was observed, particularly with indinavir and ritonavir.[29] It is important to note that glucosylceramide is considered an essential molecule in fungal biology, including in *F. pedrosoi*, and has been implicated in growth, differentiation, membrane integrity, and virulence.[24,25,104] In addition, lipid homeostasis in fungal membranes must be maintained in balance; otherwise, it can activate death pathways.[104] In this context, the increase in glucosylceramide levels observed in our results may help explain the inhibitory action of HIV-PIs on *F. pedrosoi* conidia, affecting its growth. Interestingly, ritonavir and nelfinavir also inhibited the production of phospholipase and esterase enzymes.[29] The ability of HIV-PIs to suppress non-proteolytic enzymes had previously been reported in other fungal species, for example *C. albicans*.[91]

Given that combination therapy can enhance drug efficacy while reducing toxicity,[105] we investigated the effects of HIV-PIs used in association with conventional antifungal agents at sub-inhibitory concentrations. Our results demonstrated, using spread plate technique, that combination between nelfinavir and amphotericin B reduced ~70% *F. pedrosoi* conidia viability.[28] In the same way, enhanced antifungal activity was observed for ritonavir in combination with ketoconazole and itraconazole against *P. verrucosa* conidia.[37] These findings

support the exploration of multitarget therapies or drugs with diverse mechanisms of action as strategies for treating fungal infections. The combination of two or more HIV-PIs, or their concomitant use with antifungal drugs, may alter pharmacokinetic parameters through to drug-drug interactions, potentially enhancing absorption, inhibiting drug-metabolising enzymes, and resulting in higher and more prolonged drug levels.[106]

In addition to aspartic peptidase activity, conidial cells of *P. verrucosa* and *F. pedrosoi* were shown to secrete metallopeptidases.[26,36] Concentrated culture supernatants from both fungi degraded soluble serum albumin, with *P. verrucosa* exhibiting an optimum pH of 3.0, similar to *Scedosporium (Pseudallescheria) boydii*, whose proteolytic activity is maximal under acidic conditions.[107] The enzyme secreted by *F. pedrosoi* was active ranging from acidic to alkaline conditions; although its optimum pH has not yet been determined, higher proteolytic activity was observed at the tested pH values of 5.5 and 7.2.[26] Fungal metalloproteinases can be active over a wide range of pH values; nevertheless, their activity may vary according to species and environmental conditions.[108] It is important to highlight that the enzymatic activities were strongly inhibited by chelating agents, such as phen and ethylene glycol tetraacetic acid, corroborating their classification as metallopeptidases. Moreover, we showed that *P. verrucosa* metallopeptidase activity was stimulated by the divalent cation zinc in a dose-dependent manner. Further analyses demonstrated that both enzymes secreted by these fungi could cleave human immunoglobulin G and extracellular matrix components, including human fibronectin and placental laminin.[30,36] It is worth noting that secreted metallopeptidases which belong to a zinc-dependent family known as fungalysins, have been described in some fungi including *A. fumigatus* and *C. neoformans*.[79,109,110] These enzymes also have ability to degrade the extracellular matrix proteins and are proposed as potential virulence factors for pathogenic fungi.[108]

Furthermore, our data revealed that phen inhibited *F. pedrosoi* and *P. verrucosa* conidial cells proliferation as previously described for other fungi including *Trichoderma harzianum*, *A. niger* and *P. brasiliensis*.[111,112] The effective inhibition was confirmed using antifungal susceptibility testing according to CLSI guidelines (M38-A2 document), which defined minimum inhibitory concentration (MIC) values of 3 μM (0.6 mg/L) for *F. pedrosoi* and 4 μM (0.8 mg/L) for *P. verrucosa*.[36,113] Moreover, the ability of phen to affect *P. verrucosa* viability was corroborated by the induction of marked morphological alterations, including an increase in electron-translucent vacuoles and the disruption of the outermost fibrillar layer.[36] In addition, our data showed that phen inhibited *F. pedrosoi* biofilm production and the transformation of *P. verrucosa* conidia into mycelia, at low concentrations. Our results support the idea that phen can impair key aspects of microbial metal metabolism by interfering with ion uptake and limiting their availability for essential cellular reactions. Chelating compounds are known to inhibit the activity of metal-dependent molecules, such as metalloproteins including metallopeptidases, thereby disrupting

microbial homeostasis and ultimately leading to the inhibition of critical biological processes, including nutrition, proliferation, differentiation, adhesion, invasion, dissemination, and infection (for a review, see Santos et al.[114]).

## Metal-based compounds containing 1,10-phenanthroline

Based on our findings showing that phen inhibited not only secreted enzymatic activity but also the growth and cell differentiation of *P. verrucosa* and *F. pedrosoi*,[26,36,113] we were motivated to investigate less toxic phen-derived compounds with antifungal activity. In this context, coordination compounds employing phen as a ligand, synthesised by the research group headed by Dr Malachy McCann, were selected as the focus of our studies. Phen is a bidentate chelating agent composed of a central benzene ring with two adjacent nitrogen atoms, exhibits structural features such as planarity, rigidity, and hydrophobicity, which enable it to coordinate with transition metals and form stable complexes.[114,115] For these reasons, phen is widely used in coordination chemistry, making it an attractive scaffold for drug development.[115] Consequently, metal complexes bearing phen ligands have emerged as promising candidates for therapeutic applications, owing to their diverse biological activities including antifungal properties.[116,117] McCann and coworkers demonstrated that phen-based metal complexes exhibit significant antifungal activity against *C. albicans*.[118,119,120] In particular, silver and copper complexes coordinated to 1,10-phenanthroline-5,6-dione (phendione), a ligand with O,O′-quinonoid and *N,N′*-diimine chelating functionalities, including [Ag(phendione)$_2$]ClO$_4$ (Ag-phendione) and [Cu(phendione)$_3$]$_2$·4H$_2$O (Cu-phendione), strongly inhibited yeast growth, induced morphological alterations, promoted nonspecific DNA cleavage, and interfered with cell division.[112,118,121] McCann et al.[122] also reported good tolerability of phen, phendione, and their silver and copper complexes based on *in vivo* assays in *Galleria mellonella* and Swiss mice.

Building on these findings, our group demonstrated that Ag-phendione and Cu-phendione markedly inhibited *P. verrucosa* growth and exhibited fungicidal activity.[123] To enhance antifungal activity, a series of novel phen-derived manganese(II), copper(II), and silver(I) complexes incorporating dicarboxylic acids were synthesised by McCann's research group, and thirteen of these compounds were evaluated against *F. pedrosoi*. This fungus was more sensitive to the silver(I)-complexes [Ag(phen)$_2$]ClO$_4$ (Ag-phen) and [Ag$_2$(3,6,9-trioxaundecanedioate)(phen)$_4$]·EtOH (Ag-tdda-phen).[113] Thus, this review discusses the effects of these complexes on pathophysiological processes in *P. verrucosa* and *F. pedrosoi*. Firstly, we showed that the MIC values of either free ligands phen and phendione, or metal salts (such as cooper and silver perchlorate) were higher than those observed for the metal complexes on *P. verrucosa* and *F. pedrosoi*.[113,123] These findings corroborate that the activity of the metal complexes is superior as expected and described for other organisms.[122,124] The effective antifungal action was confirmed since both Ag-phendione and Cu-phendione were able to induce ultrastructural alterations in *P. verrucosa* conidia, indicative of cell death, such as cell wall detachment, surface invaginations, cell ruptures, and shrinkage.[123] To assess the relevance of combination therapies, we evaluated Cu-phendione, which displays fungicidal activity against *P. verrucosa*, in association with amphotericin B at sub-inhibitory concentrations. Under these conditions, fungal proliferation was reduced by approximately 40%.[123] A subsequent checkerboard assay confirmed increased antifungal activity against *P. verrucosa* resulting from the combination of Cu-phendione and amphotericin B, which was classified as an additive effect.[125]

The mode of action of these phen-derived metal complexes has not been completely elucidated, but interestingly, they can exhibit different mechanisms compared to those reported for conventional antifungal agents used in clinical practice.[112,116] For fungal cells, McCann et al.[112] documented that phen-derived metal complexes can modulate several events, including the cell membrane and organelles disruption, mitochondrial function damage, essential metals sequestration, nuclear DNA degradation, and control of cell division alterations. In an effort to understand the mechanisms underlying the antifungal activity against CBM-associated fungi, our findings demonstrated that phendione and its silver(I)-complex were able to reduce *P. verrucosa* ergosterol levels, the major target for a variety of antifungal agents.[123] Our data were similar to those described for *C. albicans*, which also exhibits inhibited ergosterol production when exposed to phen-derived metal complexes.[119] As reactive oxygen species induced microbial killing has been extensively investigated as a potential mechanism of drug action, we evaluated the effects of Ag-phen and Ag-tdda-phen and demonstrated that both complexes induce reactive oxygen species production in *F. pedrosoi*.[113] In fact, reactive oxygen species are well known to oxidise biological macromolecules such as DNA, proteins, and lipids, ultimately leading to cell death, which supports the idea that oxidative stress contributes to the antifungal activity of these complexes.[126]

Our group also studied the effect of the complexes on potential virulence factors of CBM-causing fungi. The capacity for morphological transition is a key virulence-associated feature in pathogenic fungi, enabling invasion, evasion of host defences, and spread within the host, which favours infection establishment.[127,128] Thus, the action of the complexes was evaluated on the transition of *P. verrucosa* and *F. pedrosoi* conidia to filamentous form.[113,123] Treatment of *P. verrucosa* conidia with non-inhibitory concentrations of Ag-phendione and Cu-phendione resulted in inhibition of hyphal formation, as recently described for *Scedosporium apiosperma*.[129] Likewise, Ag-phen and Ag-tdda-phen inhibited the transition of *F. pedrosoi* from conidia to mycelia.[113] It is worth noting that fungal filamentation is among the earliest steps involved in the development of biofilms.[130,131] We demonstrated that *P. verrucosa* under biofilm conditions had its biomass formation and extracellular matrix production affected after treatment with Ag-phendione and Cu-phendione in sub-

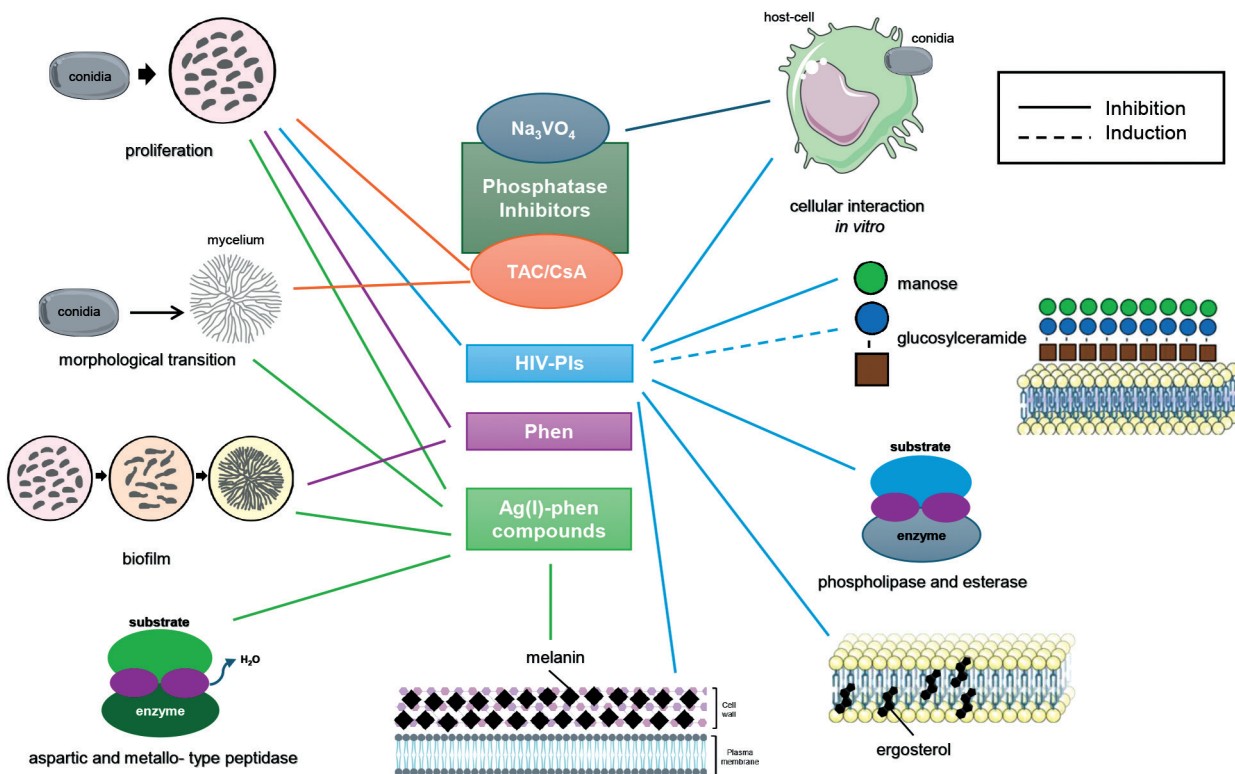

Schematic representation of the cellular targets studied for inhibitors and metal complexes active against *Fonsecaea pedrosoi*. Tacrolimus/cyclosporine A (TAC/CsA, orange) inhibited fungal proliferation and morphological transition conidia-mycelium. The acid phosphatase inhibitor sodium orthovanadate (Na₃VO₄, dark blue) affected the interaction between fungal and animal cells. Human immunodeficiency virus (HIV) peptidase inhibitors (HIV-PIs, light blue) inhibited growth, host-pathogen interaction, and cellular structures such as mannose, ergosterol, melanin, phospholipase as well as esterase, while increasing glucosylceramide levels. 1,10-phenanthroline (Phen, purple) reduced the viability of both planktonic and biofilm cells. Ag(I)-phen compounds (green) inhibited growth, filamentation, biofilm, aspartic and metallo-type peptidase activities, as well as melanin production. The figure was created using Scientific icons adapted from Bioicons (https://bioicons.com), licensed under CC BY 4.0 and Servier Medical Art, licensed under CC BY 4.0 (https://smart.servier.com).

MIC concentrations.[125] In addition, these complexes inhibited the viability of biofilm cells, showing MIC values higher than those observed for planktonic cells.[125] For *F. pedrosoi*, Ag-phen and Ag-tdda-phen were highly effective in inhibiting both biofilm-forming cells and mature biofilms.[113] Ag-phen and Ag-tdda-phen reduced the viability of mature *F. pedrosoi* biofilms by more than 90% at low concentrations, only two times and equal to the MIC values required to inhibit planktonic cells, respectively. Moreover, the extracellular matrix was inhibited by Ag-phen during biofilm formation. The efficacy of the complexes was better than itraconazole, that could not affect the viability of the *F. pedrosoi* mature biofilm. Furthermore, the combination of Ag-tdda-phen and itraconazole at non-inhibitory concentrations resulted in reduced biofilm viability and extracellular matrix content, whereas, under the same conditions, Ag-phen affected only the extracellular matrix.[113] Fungal biofilm formation contributes to increased drug resistance and commonly occurs on biotic surfaces and medical devices, representing a significant challenge for healthcare professionals.[130] Accordingly, inhibition of biofilm formation by silver(I)-phen derivatives is particularly relevant, as also described for *Candida haemulonii* and *S. apiospermum*.[129,132]

Studies have shown that metal-based drugs can inhibit enzymes by mimicking metabolic substrates or by binding to essential biomolecules, ultimately leading to cell death.[133] Considering that hydrolytic enzymes produced by *F. pedrosoi* and *P. verrucosa* are involved in key fungal biological processes, as previously discussed,[26,37] the effects of silver(I)-derived compounds on these enzymatic activities were evaluated. Metallopeptidase activity secreted by *F. pedrosoi* and *P. verrucosa* was inhibited by both phen and phendione complexes, respectively.[113,123] We also observed that treatment with Ag-phen and Ag-tdda-phen reduced the activity of the aspartic peptidase secreted by *F. pedrosoi*.[113] However, under the conditions employed, none of the complexes exhibited any effect on *F. pedrosoi* ectophosphatase as well as esterase and phospholipase activities detected previously by our group.[31] In addition, the effect of the complexes on *F. pedrosoi* melanin, a key virulence factor in dematiaceous fungi, was assessed. Interestingly, our data demonstrated that *F. pedrosoi* cells treated with non-inhibitory concentrations of Ag-phen and Ag-tdda-phen showed reduced recognition by the anti-melanin monoclonal antibody 6D2, suggesting inhibition of melanin production.[113] Thus, compounds that inhibit melanin may offer alternative therapeutic strategies for the management of CBM.

Considering the crucial role of fungus-host cell interactions in CBM-associated fungal pathogenesis,[5,102] we investigated the effects of the complexes on *P. verrucosa* conidia after their interaction with human macrophages derived from THP-1 cells. Our results showed that both pre- and post-treatment of *P. verrucosa* conidia with Ag-phendione significantly reduced intracellular conidial viability.[125] Based on these findings and the lower toxicity of Ag-phendione to THP-1 cells, this complex was selected for efficacy evaluation in *in vivo* assays using *G. mellonella* larvae.[125] We demonstrated that Ag-phendione increased larval survival, indicating a protective effect in *G. mellonella* larvae infected with *P. verrucosa* conidia.[125] Our findings are in agreement with previous studies involving *C. haemulonii* and with more recent reports on *C. neoformans*, which also showed reduced *G. mellonella* mortality following treatment with phen- and phendione-based metal complexes. [134,135] These results reinforce the therapeutic potential of phen/phendione-derived metal complexes, as they have demonstrated efficacy in both cellular systems and *in vivo* models of fungal infection. Additional experiments are currently being conducted by our group to evaluate the efficacy of phen-based complexes using *in silico*, *in vitro* and *in vivo* approaches aiming to identify compounds with selective action and reduced toxicity against CBM-causing fungi.

## In Conclusions

Over the years, our group has conducted extensive research on CBM-causing fungi, particularly *F. pedrosoi*, providing early descriptions of aspartic and metallopeptidases, cell wall-associated phosphatases (ectophosphatases), intracellular phosphatases (such as calcineurin), and biofilm formation. Our data revealed that these enzymes and structures are critical for the biology and/or virulence of CBM-associated fungi and represent promising antifungal targets, since phosphatase and peptidase inhibitors, along with phen-derived compounds, effectively modulated key processes including fungal proliferation, filamentation, and cellular interactions (Figure). These findings underscore the importance of developing selective inhibitors or compounds that specifically target fungal cells, enabling lower dosages and minimising toxicity. Current studies are incorporating a broader spectrum of clinical isolates of CBM-causing fungi, aiming to better elucidate the mechanisms through which these inhibitors and compounds exert their effects.

## ACKNOWLEDGEMENTS

To all co-authors and collaborators involved in the studies published by the group and cited in this review, especially Drs Celuta Sales Alviano (*In memoriam*), José Roberto Meyer-Fernandes, Luiz Rodolpho Raja Gabaglia Travassos (*In memoriam*), Malachy McCann, and Michael Devereux. The authors would like to dedicate this review to Professor Celuta Sales Alviano, whose direct and indirect support made all these studies possible and whose enormous contribution to chromoblastomycosis research is deeply appreciated.

## AUTHORS' CONTRIBUTION

LFK conceived and designed the review; ISS, MQG, VFP and LFK conducted the writing-original draft preparation of the manuscript; ISS, ALS and LFK writing-review, editing and revision and prepared figures and tables. The funding agencies were not involved in the design of the study, collection and analysis of data, decision to publish, or writing of the manuscript. None of the authors has a conflict of interest to disclose.

## DATA AVAILABILITY

The contents underlying the research text are included in the manuscript.

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

# OPEN PEER REVIEW

Memórias do IOC thanks the anonymous reviewers for their contribution to the peer review of this work.

**FIRST REVIEW ROUND**

REVIEWERS' COMMENTS

**REVIEWER #1**

The review by Sousa e Souza et al. is relevant to the researchers studying fungal species causing human mycosis, especially subcutaneous infections caused by Fonsecaea pedrosoi. The work critically reviews the data, collected over many years by the group, on phosphatases and peptidases and their inhibitors that have potential antifungal activity. The review is timely and welcome, however extensive revision related to the format is needed to improve readability and understanding.

Specific comments:
1. Please carry out a complete text revision for grammar, use of commas, hyphens (e.g. L. 189), style, wrong use of "regarding" and other words. Some suggestions are below, but there are a number of other sentences to polish. In several instances, the ideas are incomplete because words are missing.
2. Please consider using CBM-causing (or -associated, as in the abstract) fungi instead of CBM fungi. As a short title, chromoblastomycosis-causing fungi would be better for the readers. Please avoid the abbreviation in the acknowledgment. Also consider restricting the abbreviations to CBM (which this reviewer considers not necessary either) due to the fact that a text full of abbreviations is unpleasant to follow (e.g., NTD, phen...unnecessary). In figure 1, abbreviations are welcome, but they have to be specified in the legend.
3. 81-94: the paragraph needs some polishing. Besides repeating the CBM abbreviation and doubling "for several years", the studies on Fonsecaea pedrosoi have to be better introduced before the peptidases are mentioned in line 90, as if they had been described before in the text. "Several" should be "many". Maybe you could explain in this paragraph that there are few works with the same focus coming from other groups that study this group of fungi in order to justify a review centered on the works reported by the group.
L. 118-119: Please revise spelling and the need of commas.
L. 130: Co2+ or Ca2+?l.164: revise the conclusion.
L. 184: please complete the idea. Target of what?L. 207: Candidozyma auris.
L. 303: "...sclerotic cell culture supernatants did not degrade fibronectin.
L. 304: "culture supernatant" missing.
L. 304-307: please polish for clear understanding.
L. 330-331: rephrase.
L. 358-359; L. 368-369: what would be the mechanism involved in this effect?
L. 366: ...integrity of the fungal plasma membraneL. 386-388: please rephrase.
L. 402-405;
L. 426-427: Does an optimum pH range from 2,0 -10,0 makes sense for one sole group of proteinases? Considering that 1,10 phenanthroline can affect other cell functions, as pointed out later in the text, these conclusions should be taken more critically and not focusing only on the metalloproteinases.
Figure 1: maybe you could use bigger fonts.
Table 1 legend: please rephrase.

AUTHORS' RESPONSE TO THE REVIEWERS

Dear Editor,
I appreciate your message concerning the Manuscript ID MIOC-2025-0305 entitled "Novel targets and potential therapeutic approaches for chromoblastomycosis: phosphatase/peptidase inhibitors and metal-based compounds containing 1,10-phenanthroline" submitted for publication in Memórias do Instituto Oswaldo Cruz, as a review article for the Special Issue "Commemorative article series celebrating the 125th anniversary of the IOC", series 1 (Cellular and Molecular Biology, Diagnosis, Therapies, and Biomodels). We are currently resubmitting our review article following your proposals as a revised manuscript. Bearing this in mind, I would like to express our position concerning the points raised by the reviewer. Eventually, the authors hope that the new manuscript version is appropriate to be published in this renowned journal.

Reviewer 1 comments:

The review by Sousa e Souza et al. is relevant to the researchers studying fungal species causing human mycosis, especially subcutaneous infections caused by Fonsecaea pedrosoi. The work critically reviews the data, collected over many years by the group, on phosphatases and peptidases and their inhibitors that have potential antifungal activity. The review is timely and welcome, however extensive revision related to the format is needed to improve readability and understanding.

Authors: The authors sincerely thank the reviewer for the positive comments and are certain that addressing the reviewer's suggestions has significantly improved the quality of this review article.

Specific comments:

1. Please carry out a complete text revision for grammar, use of commas, hyphens (e.g. L. 189), style, wrong use of "regarding" and other words. Some suggestions are below, but there are a number of other sentences to polish. In several instances, the ideas are incomplete because words are missing.

Authors: The text was thoroughly revised following the reviewer's recommendations.

2. Please consider using CBM-causing (or -associated, as in the abstract) fungi instead of CBM fungi. As a short title, chromoblastomycosis-causing fungi would be better for the readers. Please avoid the abbreviation in the acknowledgment. Also consider restricting the abbreviations to CBM (which this reviewer considers not necessary either) due to the fact that a text full of abbreviations is unpleasant to follow (e.g., NTD, phen...unnecessary). In figure 1, abbreviations are welcome, but they have to be specified in the legend.

Authors: All changes suggested by the reviewer have been implemented, except for the restriction on the abbreviations "CBM" (chromoblastomycosis) and "phen" (1,10-phenanthroline). These terms are used repeatedly throughout the manuscript, and the latter is required for the abbreviations of the compounds Ag–phen and Ag–tdda–phen. As suggested by the reviewer, several abbreviations, including NTD, WHO, Pi, p-NPP, CLSI, FK506, CsA, EGTA, Ca2+, Zn2+, Co2+, Fe3+, FDA, HAART, AIDS, AMB and ROS, were removed from the manuscript. In addition, the abbreviations from Figure 1 were also provided in this figure legend.

3. 81-94: the paragraph needs some polishing. Besides repeating the CBM abbreviation and doubling "for several years", the studies on Fonsecaea pedrosoi have to be better introduced before the peptidases are mentioned in line 90, as if they had been described before in the text. "Several" should be "many". Maybe you could explain in this paragraph that there are few works with the same focus coming from other groups that study this group of fungi in order to justify a review centered on the works reported by the group.

Authors: The authors revised the paragraph and added a sentence to explain the rationale for focusing the review on the works reported by the group (see revised text below). It is also important to mention that this review article was submitted for publication in Memórias do Instituto Oswaldo Cruz as part of the Special Issue "Commemorative article series celebrating the 125th anniversary of the IOC". In this context, the review was conceived to highlight our group's contributions to the study of chromoblastomycosis over recent years, within the broader framework of advances in this field.

Studies from our group, derived from a sustained line of investigation, have contributed to the understanding of key cellular and biological events in CBM-causing fungi. These findings form the basis of the present review, which integrates them into a unified perspective and provides a framework to guide future studies, focusing on the identification of novel targets and potential therapeutic strategies for this neglected and debilitating infection.

L. 118-119: Please revise spelling and the need of commas.

Authors: The paragraph was thoroughly revised.

L. 130: Co2+ or Ca2+? L.164: revise the conclusion.

Authors: The sentence is correct. The acid phosphatase of R. aquaspersa conidia was stimulated by cobalt. In order to avoid any misunderstanding, the chemical formulas were replaced by the corresponding ion names written in full. The paragraph including the original L.164 was revised and is correct.

L. 184: please complete the idea. Target of what?

Authors: We changed "...a potential target..." to "...a potential antifungal target...".

L. 207: Candidozyma auris.

Authors: The name Candidozyma was inserted.

L. 303: "...sclerotic cell culture supernatants did not degrade fibronectin.

Authors: In order to improve clarity, the authors repeated the word "fibronectin" at the end of the sentence, as suggested by the reviewer.

L. 304: "culture supernatant" missing.
Authors: The sentence "…P. verrucosa conidia hydrolyzed…" was changed to "…the culture supernatant of P. verrucosa conidia hydrolyzed…"

L. 304-307: please polish for clear understanding.
Authors: The paragraph was revised and is now clear.

In line with these observations, the culture supernatant of P. verrucosa conidia hydrolyzed human serum albumin, and aspartic peptidase activity in both fungi was supported by the substantial reduction in enzymatic activity following treatment with the classical inhibitor pepstatin A and HIV-PIs.

L. 330-331: rephrase.
Authors: The text was rephrased as recommended by the reviewer.

As the ability to survive and replicate within host cells is a key determinant of fungal virulence (127), we examined the impact of HIV-PIs on the adhesion of F. pedrosoi and P. verrucosa conidia to animal cells.

L. 358-359; and L. 402-405;
Authors: We are sorry for any misunderstanding, but we did not find an appropriate use for a semicolon in these sentences.

L. 368-369: what would be the mechanism involved in this effect?
Authors: Although the precise mechanism of action remains unclear, the antifungal activity of the HIV protease inhibitors mentioned in this paragraph may be mediated, at least in part, by ergosterol depletion, which could indirectly impair membrane-associated processes and other virulence-related pathways. Part of this sentence was included in the revised manuscript to strengthen and clarify the discussion.

L. 366: ...integrity of the fungal plasma membraneL.
Authors: The missing information 'of the fungal plasma membrane' was added.

L 386-388: please rephrase.
Authors: The sentence was rephrased as suggested by the reviewer to improve clarity and grammatical accuracy.

Given that combination therapy can enhance drug efficacy while reducing toxicity (105), we investigated the effects of HIV-PIs used in association with conventional antifungal agents at sub-inhibitory concentrations.

L. 426-427: Does an optimum pH range from 2,0 -10,0 makes sense for one sole group of proteinases? Considering that 1,10 phenanthroline can affect other cell functions, as pointed out later in the text, these conclusions should be taken more critically and not focusing only on the metalloproteinases.
Authors: The authors would like to thank the reviewer for the comments, and the text was adjusted in the revised manuscript to clarify and improve the discussion. The enzyme secreted by F. pedrosoi was active ranging from acidic to alkaline conditions; although, its optimum pH has not yet been determined, higher proteolytic activity was observed at the tested pH values of 5.5 and 7.2 (Palmeira et al., 2006, FEMS Immunol Med Microbiol. 2006;46(1):21–29). Fungal metalloproteinases can be active over a wide range of pH values; nevertheless, their activity may vary according to species and environmental conditions (Monod et al., 2002, Int J Med Microbiol. 2002; 292:405–419). It is important to highlight that the enzymatic activities were strongly inhibited by chelating agents, such as 1,10-phenanthroline and EGTA, corroborating their classification as metallopeptidases. Part of this text has been incorporated into the revised manuscript.

The authors have also revised the Conclusions (see revised text below), which are no longer focused exclusively on metalloproteinases.
Our results support the idea that phen can impair key aspects of microbial metal metabolism by interfering with ion uptake and limiting their availability for essential cellular reactions. Chelating compounds are known to inhibit the activity of metal-dependent molecules, such as metalloproteins including metallopeptidases, thereby disrupting microbial homeostasis and ultimately leading to the inhibition of critical biological processes, including nutrition, proliferation, differentiation, adhesion, invasion, dissemination, and infection (for a review, see Santos et al. (114)).

Figure 1: maybe you could use bigger fonts.
Authors: In the revised version, the quality of Figure 1 was improved, resulting in clearer fonts.

Table 1 legend: please rephrase.
Authors: We rephrased as recommended by the reviewer.

## SECOND REVIEW ROUND

**REVIEWERS' COMMENTS**

### REVIEWER #1

No comments.

