## [Reviewer Report · FIRST REVIEW ROUND - REVIEWERS COMMENTS]

## REVIEWER #1

The review by Sousa e Souza et al. is relevant to the researchers studying fungal species causing human mycosis, especially subcutaneous infections caused by *Fonsecaea pedrosoi*. The work critically reviews the data, collected over many years by the group, on phosphatases and peptidases and their inhibitors that have potential antifungal activity. The review is timely and welcome, however extensive revision related to the format is needed to improve readability and understanding.

**Specific comments:**

1. Please carry out a complete text revision for grammar, use of commas, hyphens (e.g. L. 189), style, wrong use of “regarding” and other words. Some suggestions are below, but there are a number of other sentences to polish. In several instances, the ideas are incomplete because words are missing.

2. Please consider using CBM-causing (or -associated, as in the abstract) fungi instead of CBM fungi. As a short title, chromoblastomycosis-causing fungi would be better for the readers. Please avoid the abbreviation in the acknowledgment. Also consider restricting the abbreviations to CBM (which this reviewer considers not necessary either) due to the fact that a text full of abbreviations is unpleasant to follow (e.g., NTD, phen...unnecessary). In figure 1, abbreviations are welcome, but they have to be specified in the legend.

3. 81-94: the paragraph needs some polishing. Besides repeating the CBM abbreviation and doubling “for several years”, the studies on *Fonsecaea pedrosoi* have to be better introduced before the peptidases are mentioned in line 90, as if they had been described before in the text. “Several” should be “many”. Maybe you could explain in this paragraph that there are few works with the same focus coming from other groups that study this group of fungi in order to justify a review centered on the works reported by the group.

L. 118-119: Please revise spelling and the need of commas.

L. 130: Co2+ or Ca2+?l.164: revise the conclusion.

L. 184: please complete the idea. Target of what?L. 207: *Candidozyma auris*.

L. 303: “...sclerotic cell culture supernatants did not degrade fibronectin.

L. 304: “culture supernatant” missing.

L. 304-307: please polish for clear understanding.

L. 330-331: rephrase.

L. 358-359; L. 368-369: what would be the mechanism involved in this effect?

L. 366: ...integrity of the fungal plasma membraneL. 386-388: please rephrase.

L. 402-405; L. 426-427: Does an optimum pH range from 2,0 -10,0 makes sense for one sole group of proteinases? Considering that 1,10 phenanthroline can affect other cell functions, as pointed out later in the text, these conclusions should be taken more critically and not focusing only on the metalloproteinases.

Figure 1: maybe you could use bigger fonts.

Table 1 legend: please rephrase.

## AUTHORS’ RESPONSE TO THE REVIEWERS

Dear Editor,

I appreciate your message concerning the Manuscript ID MIOC-2025-0305 entitled “Novel targets and potential therapeutic approaches for chromoblastomycosis: phosphatase/peptidase inhibitors and metal-based compounds containing 1,10-phenanthroline” submitted for publication in Memórias do Instituto Oswaldo Cruz, as a review article for the Special Issue “Commemorative article series celebrating the 125th anniversary of the IOC”, series 1 (Cellular and Molecular Biology, Diagnosis, Therapies, and Biomodels).

We are currently resubmitting our review article following your proposals as a revised manuscript. Bearing this in mind, I would like to express our position concerning the points raised by the reviewer. Eventually, the authors hope that the new manuscript version is appropriate to be published in this renowned journal.

**Reviewer 1 comments:**

*The review by Sousa e Souza et al. is relevant to the researchers studying fungal species causing human mycosis, especially subcutaneous infections caused by Fonsecaea pedrosoi. The work critically reviews the data, collected over many years by the group, on phosphatases and peptidases and their inhibitors that have potential antifungal activity. The review is timely and welcome, however extensive revision related to the format is needed to improve readability and understanding.*

**Authors:** The authors sincerely thank the reviewer for the positive comments and are certain that addressing the reviewer’s suggestions has significantly improved the quality of this review article.

**Specific comments:**

*1. Please carry out a complete text revision for grammar, use of commas, hyphens (e.g. L. 189), style, wrong use of “regarding” and other words. Some suggestions are below, but there are a number of other sentences to polish. In several instances, the ideas are incomplete because words are missing.*

**Authors:** The text was thoroughly revised following the reviewer’s recommendations.

*2. Please consider using CBM-causing (or -associated, as in the abstract) fungi instead of CBM fungi. As a short title, chromoblastomycosis-causing fungi would be better for the readers. Please avoid the abbreviation in the acknowledgment. Also consider restricting the abbreviations to CBM (which this reviewer considers not necessary either) due to the fact that a text full of abbreviations is unpleasant to follow (e.g., NTD, phen...unnecessary). In figure 1, abbreviations are welcome, but they have to be specified in the legend.*

**Authors:** All changes suggested by the reviewer have been implemented, except for the restriction on the abbreviations “CBM” (chromoblastomycosis) and “phen” (1,10-phenanthroline). These terms are used repeatedly throughout the manuscript, and the latter is required for the abbreviations of the compounds Ag–phen and Ag–tdda–phen.

As suggested by the reviewer, several abbreviations, including NTD, WHO, Pi, *p*-NPP, CLSI, FK506, CsA, EGTA, Ca2+, Zn2+, Co2+, Fe3+, FDA, HAART, AIDS, AMB and ROS, were removed from the manuscript.

In addition, the abbreviations from Figure 1 were also provided in this figure legend.

*3. 81-94: the paragraph needs some polishing. Besides repeating the CBM abbreviation and doubling “for several years”, the studies on Fonsecaea pedrosoi have to be better introduced before the peptidases are mentioned in line 90, as if they had been described before in the text. “Several” should be “many”. Maybe you could explain in this paragraph that there are few works with the same focus coming from other groups that study this group of fungi in order to justify a review centered on the works reported by the group.*

**Authors:** The authors revised the paragraph and added a sentence to explain the rationale for focusing the review on the works reported by the group (see revised text below).

It is also important to mention that this review article was submitted for publication in Memórias do Instituto Oswaldo Cruz as part of the Special Issue “Commemorative article series celebrating the 125th anniversary of the IOC”. In this context, the review was conceived to highlight our group’s contributions to the study of chromoblastomycosis over recent years, within the broader framework of advances in this field.

Studies from our group, derived from a sustained line of investigation, have contributed to the understanding of key cellular and biological events in CBM-causing fungi. These findings form the basis of the present review, which integrates them into a unified perspective and provides a framework to guide future studies, focusing on the identification of novel targets and potential therapeutic strategies for this neglected and debilitating infection.

*L. 118-119: Please revise spelling and the need of commas.*

**Authors:** The paragraph was thoroughly revised.

*L. 130: Co2+ or Ca2+? L.164: revise the conclusion.*

**Authors:** The sentence is correct. The acid phosphatase of *R. aquaspersa* conidia was stimulated by cobalt. In order to avoid any misunderstanding, the chemical formulas were replaced by the corresponding ion names written in full.

The paragraph including the original L.164 was revised and is correct.

*L. 184: please complete the idea. Target of what?*

**Authors:** We changed “…a potential target…” to “…a potential antifungal target…”.

*L. 207: Candidozyma auris.*

**Authors:** The name *Candidozyma* was inserted.

*L. 303: “...sclerotic cell culture supernatants did not degrade fibronectin.*

**Authors:** In order to improve clarity, the authors repeated the word “fibronectin” at the end of the sentence, as suggested by the reviewer.

*L. 304: “culture supernatant” missing.*

**Authors:** The sentence “…*P. verrucosa* conidia hydrolyzed…” was changed to “…the culture supernatant of *P. verrucosa* conidia hydrolyzed…”

*L. 304-307: please polish for clear understanding.*

**Authors:** The paragraph was revised and is now clear.

In line with these observations, the culture supernatant of *P. verrucosa* conidia hydrolyzed human serum albumin, and aspartic peptidase activity in both fungi was supported by the substantial reduction in enzymatic activity following treatment with the classical inhibitor pepstatin A and HIV-PIs.

*L. 330-331: rephrase.*

**Authors:** The text was rephrased as recommended by the reviewer.

As the ability to survive and replicate within host cells is a key determinant of fungal virulence (127), we examined the impact of HIV-PIs on the adhesion of *F. pedrosoi* and *P. verrucosa* conidia to animal cells.

*L. 358-359; and L. 402-405;*

**Authors:** We are sorry for any misunderstanding, but we did not find an appropriate use for a semicolon in these sentences.

*L. 368-369: what would be the mechanism involved in this effect?*

**Authors:** Although the precise mechanism of action remains unclear, the antifungal activity of the HIV protease inhibitors mentioned in this paragraph may be mediated, at least in part, by ergosterol depletion, which could indirectly impair membrane-associated processes and other virulence-related pathways.

Part of this sentence was included in the revised manuscript to strengthen and clarify the discussion.

*L. 366: ...integrity of the fungal plasma membraneL.*

**Authors:** The missing information ‘of the fungal plasma membrane’ was added.

*L 386-388: please rephrase.*

**Authors:** The sentence was rephrased as suggested by the reviewer to improve clarity and grammatical accuracy.

Given that combination therapy can enhance drug efficacy while reducing toxicity (105), we investigated the effects of HIV-PIs used in association with conventional antifungal agents at sub-inhibitory concentrations.

*L. 426-427: Does an optimum pH range from 2,0 -10,0 makes sense for one sole group of proteinases?*

*Considering that 1,10 phenanthroline can affect other cell functions, as pointed out later in the text, these conclusions should be taken more critically and not focusing only on the metalloproteinases.*

**Authors:** The authors would like to thank the reviewer for the comments, and the text was adjusted in the revised manuscript to clarify and improve the discussion.

The enzyme secreted by *F. pedrosoi* was active ranging from acidic to alkaline conditions; although, its optimum pH has not yet been determined, higher proteolytic activity was observed at the tested pH values of 5.5 and 7.2 (Palmeira et al., 2006, FEMS Immunol Med Microbiol. 2006;46(1):21–29).

Fungal metalloproteinases can be active over a wide range of pH values; nevertheless, their activity may vary according to species and environmental conditions (Monod et al., 2002, Int J Med Microbiol. 2002; 292:405–419).

It is important to highlight that the enzymatic activities were strongly inhibited by chelating agents, such as 1,10-phenanthroline and EGTA, corroborating their classification as metallopeptidases.

Part of this text has been incorporated into the revised manuscript.

The authors have also revised the Conclusions (see revised text below), which are no longer focused exclusively on metalloproteinases.

Our results support the idea that phen can impair key aspects of microbial metal metabolism by interfering with ion uptake and limiting their availability for essential cellular reactions.

Chelating compounds are known to inhibit the activity of metal-dependent molecules, such as metalloproteins including metallopeptidases, thereby disrupting microbial homeostasis and ultimately leading to the inhibition of critical biological processes, including nutrition, proliferation, differentiation, adhesion, invasion, dissemination, and infection (for a review, see Santos et al. (114)).

*Figure 1: maybe you could use bigger fonts.*

**Authors:** In the revised version, the quality of Figure 1 was improved, resulting in clearer fonts.

*Table 1 legend: please rephrase.*

**Authors:** We rephrased as recommended by the reviewer.